# Antibiotic Treatment for Lower Respiratory Tract Infections in Primary Care: A Register-Based Study Examining the Role of Radiographic Imaging

**DOI:** 10.3390/antibiotics12071165

**Published:** 2023-07-09

**Authors:** Sara Carlsson, Katarina Hedin, Olof Cronberg, Anna Moberg

**Affiliations:** 1Department of Infection Disease and Control, Region Östergötland, 58185 Linköping, Sweden; sara.a.carlsson@regionostergotland.se; 2Futurum—The Academy for Health and Care, Region Jönköping County, 55185 Jönköping, Sweden; katarina.hedin@rjl.se; 3Department of Clinical Sciences in Malmö, Family Medicine, Lund University, 20502 Malmö, Sweden; olof.cronberg@kronoberg.se; 4Department of Health, Medicine and Caring Sciences, Linköping University, 58183 Linköping, Sweden; 5Växjöhälsan Primary Healthcare Center and Department of Research and Development, Region Kronoberg, 35112 Växjö, Sweden; 6Kärna Primary Healthcare Center, Region Östergötland, 58662 Linköping, Sweden

**Keywords:** primary care, lower respiratory tract infections, pneumonia, acute bronchitis, chest imaging, characteristics

## Abstract

When imaging (i.e., chest-x-ray or computed tomography) is used to differentiate between acute bronchitis and pneumonia, many patients are being prescribed antibiotics despite the absence of radiographic pneumonia signs. This study of lower respiratory tract infections (LRTIs) with negative chest imaging compares cases where antibiotics were prescribed and not prescribed to find characteristics that could explain the prescription. Data were extracted from the regional electronic medical record system in Kronoberg County, Sweden, for patients aged 18–79 years diagnosed with acute bronchitis or pneumonia and who had any chest radiologic imaging between 2007–2014. Of 696 cases without evidence of pneumonia on imaging, 55% were prescribed antibiotics. Age, sex, and co-morbidity did not differ between those with or without antibiotics. The median level of C-reactive protein was low in both groups but differed significantly (21 vs. 10 mg/L; *p* < 0.001). Resident physicians prescribed antibiotics more frequently than interns or specialists (*p* < 0.001). It is unclear what features prompted the antibiotic prescribing in those with negative imaging indicating overuse of antibiotics for LRTIs.

## 1. Introduction

Respiratory tract infections (RTIs) are common in primary care. Lower respiratory tract infections (LRTIs), including pneumonia, acute bronchitis, and acute exacerbation of chronic obstructive pulmonary disease (COPD), account for about 25% of all RTIs [1]. Although important, it can be challenging to differentiate between pneumonia and acute bronchitis [2]. Pneumonia is a serious infection, often caused by bacteria, and should be treated with antibiotics. In Sweden, phenoxymethylpenicillin, PcV, is the first-line antibiotic therapy for pneumonia. Acute bronchitis, on the other hand, is most often of viral origin and a self-limiting disease without a need for antibiotics [3,4].

Among nations worldwide, Sweden has low rates of antibiotic consumption and antimicrobial resistance [5]. Inappropriate antibiotic use contributes to the emergence of antimicrobial resistance, which is increasing worldwide, causing at least 700,000 deaths annually, and is one of the biggest threats to global health [6]. Antibiotics should therefore be used only when patients benefit from such treatment.

Identifying pneumonia in primary care is often complex, especially in moderately ill patients [7]. Efforts to formulate clinical decision tools to help physicians to rule in or rule out the diagnosis of pneumonia have had varying degrees of success [4,8,9,10,11]. The gold standard for pneumonia diagnosis is chest X-ray (CXR) [9,12]. However, imaging access in primary care is almost always limited and rarely crucial for diagnosis. The Swedish guideline for primary care suggests CXR be considered only when the physician is uncertain of the diagnosis after the analysis of C-reactive protein (CRP) [3]. 

Usually, clinical history, physical examination, the physician’s previous experience, and sometimes laboratory tests sum up to the diagnosis of pneumonia in primary care [8,9]. This approach often results in over-diagnosing of pneumonia and, thus, overuse of antibiotics, although under-diagnosing of pneumonia has also been reported [3]. Several studies have shown that when CXR is used to differentiate between acute bronchitis and pneumonia, a high proportion of the patients are prescribed antibiotics and also when typical signs of pneumonia are absent. The reasons why these patients are prescribed antibiotics are not known [13,14].

The primary objective of this study was to look at the pattern and prevalence of antibiotic prescribing in cases of acute bronchitis or pneumonia when chest imaging was negative. The aim was also to compare characteristics between those prescribed antibiotics and those not prescribed antibiotics after a negative imaging result to identify characteristics that could predict whether or not antibiotics were prescribed after negative imaging. The secondary objective was to compare the characteristics of cases with prescribed antibiotics and a negative imaging result to cases with prescribed antibiotics and a positive imaging result to find explanations for why antibiotic treatment was prescribed even though imaging was negative.

## 2. Materials and Methods

### 2.1. Design and Study Population

This retrospective registered-based study used data from the Kronoberg infection database in primary care (KIDPC) that included information on subjects with infectious disease diagnoses and the prescribed antibiotics in the primary care of Kronoberg County, Sweden, between 2006 and 2014. In total, 33 primary health care centers (PHCCs) and three out-of-hours offices provided data. All visits with a physician require that the physician registers a diagnosis code according to the 10th revision of the International Statistical Classification of Diseases and Related Health Problems (ICD-10) or its modified Swedish PHC edition (KSH97-*p*). The data in the KIDPC was extracted from the EMRs of Kronoberg County (Cambio Cosmic software, Cambio Healthcare Systems AB, Linköping, Sweden) on one occasion, in 2015, using Business Objects (SAP AG, Walldorf, Germany) [15].

### 2.2. Data Extraction

We included patients 18–79 years of age with acute bronchitis or pneumonia and chest imaging within 7 days after the first consultation. From the KIDPC, we extracted information on age, sex, diagnosis, CRP, and chest imaging and their results as well as information on antibiotic prescriptions and the number of antibiotic prescriptions per episode, co-morbidities (diagnosis of asthma, COPD, heart failure, diabetes (type 1 or 2), any psychiatric diagnosis (all ICD-10 codes beginning with “F”), hypertension or ischemic heart disease at any time during 2007–2014) and the consulted physician’s level of competence.

We did not include patients over 79 years of age due to the lack of EMR-connected prescription history [16]. Contacts occurring within 6 weeks for the same patient and diagnosis were defined as one episode or case. Twenty-three patients had more than one case of LRTI during the study period (2007–2014), and no one had more than two cases. The chest imaging results were divided into two groups: negative/normal examination (no signs of pneumonia) vs. positive examination (pneumonia signs visible).

### 2.3. Statistical Methods

Proportions and medians were calculated for descriptive data. Pearson’s χ^2^ test was used when analyzing differences in proportions. Mann–Whitney U test was used to identify differences between independent samples with a skewed distribution. Continuous variables with non-normal distribution are presented as medians (interquartile range, IQR). Binary logistic regression was used when analyzing correlations between characteristics (age, sex, co-morbidity, median CRP, level of prescriber competence) and antibiotic treatment. All statistical analyses were performed using IBM SPSS Statistics (version 27). *p* values < 0.05 were considered significant. Odds ratios (OR) were calculated with 95% confidence intervals [CI].

## 3. Results

### 3.1. General Characteristics

During the study period, there were 12,348 cases of pneumonia and 25,228 cases of acute bronchitis. Imaging (CXR or computed tomography) was performed in 1113 cases. In all but one of the 1113 cases, CXR was carried out. No consolidation was seen in 696 cases, Figure 1.

Characteristics of the study population are shown in Table 1. Age and sex differences were insignificant between the positive and negative imaging group. The proportion of COPD was 14% and did not differ significantly between negative and positive imaging cases (*p* = 0.76). In contrast, co-morbidities like ischemic heart disease and asthma were more common in the negative imaging group (*p* < 0.05). CRP testing was performed in 80% of the cases, and microbiological testing in 10%. There were no differences in testing rates between those with negative and positive imaging (*p* = 0.73 and *p* = 0.31, respectively). The median CRP value was significantly lower in the group with negative imaging (Table 1). No differences were seen regarding the physicians’ level of competence between the groups. Two-thirds of the patients were assessed by a specialist in family medicine, 27% by a resident physician specializing in family medicine, and 8% by an intern (during the first two years of postgraduate training) (*p* = 0.56).

### 3.2. Antibiotic Prescribing in Cases with Negative Imaging

Antibiotics were prescribed to 55% (384/696) of subjects with negative imaging (Table 1). The adjusted OR (aOR) for antibiotic prescribing when CRP > 50 mg/L was 1.9 (CI 1.2–2.8). Resident physicians prescribed antibiotics more often than specialists aOR 2.1 (CI 1.4–3.2) but not than interns aOR 1.2 (CI 0.67–2.3). Age (aOR 1.3 (CI 0.91–2.0)), sex (aOR 1.2 (CI 0.81–1.6)), and co-morbidity (aOR 0.93 (CI 0.65–1.3)) did not impact the antibiotic prescribing pattern.

CRP test was used in 82% of the cases where an antibiotic was prescribed, compared to 79% of those not prescribed antibiotics (*p* = 0.44). The median CRP value was higher among cases with antibiotics (Table 2). In patients prescribed antibiotics, CRP levels were more often above 50 mg/L than in patients with no antibiotics.

The distribution of CRP levels (below or above 50 mg/L) in cases with negative chest imaging in which antibiotics were prescribed is presented in Figure 2.

The proportion of prescribed PcV was 23% (90/384) when imaging was negative. No significant difference was seen regarding the physician’s level of competence (*p* = 0.65). Co-morbidity did not affect whether or not they were treated with PcV (*p* = 0.086).

### 3.3. All cases with Antibiotic Prescribing

Among those who were prescribed antibiotics, no differences regarding age or sex were seen between those with negative and positive imaging results (Table 3). CRP was used in 82% of the cases with negative imaging, compared to 80% in cases with positive imaging (*p* = 0.62), and the median CRP differed significantly.

Fewer prescriptions of PcV were seen in those with negative imaging. However, if a co-morbidity was present, PcV was used to the same extent in both groups (20%, *p* = 0.87). More than one prescription of antibiotics was seen in 12% of cases with negative imaging compared to 17% in cases with positive imaging (*p* = 0.11). No significant difference was seen regarding the level of prescriber competence (*p* = 0.28), and the use of PcV did not differ between the groups of prescribers (*p* = 0.42).

## 4. Discussion

### 4.1. Summary

This register-based study showed that every other patient with acute bronchitis or pneumonia and negative imaging received antibiotics despite low median CRP values. There was a slight difference in the characteristics of those who received antibiotics vs. those who did not (primary objective). The median CRP level differed significantly but was nevertheless low in both groups. Resident physicians prescribed antibiotics to two out of three patients despite negative imaging.

When comparing the characteristics of patients prescribed antibiotics after negative chest imaging with those prescribed antibiotics after positive chest imaging (secondary objective), both groups had similar distributions of age and sex. However, the median CRP level was significantly lower in the group with negative imaging.

### 4.2. Strengths and Limitations

A strength of our study was the large sample size since data from the entire county and all PHCCs and out-of-hour offices were included. Another strength was that no telehealth service was available, meaning every outpatient episode was included.

Our study had a few limitations that may have impacted the results. Data on smoking, concomitant medications, allergies, and immune system (competent or compromised) were missing. In addition, the timing of the antibiotic prescribing in relation to the timing of the radiologic procedure was not known. Some prescriptions might have been canceled when the imaging result was negative. Moreover, only prescriptions made in the EMRs are included. Some prescriptions are missing due to being directly prescribed in a dose-dispensing system without connection to the EMRs. Thus, potentially impacting data from patients ≥ 75 years old.

### 4.3. Comparison with Existing Literature

In our study, 55% of the LRTI-cases with negative chest imaging were prescribed antibiotics. This is a larger proportion than in former studies, where the proportion of antibiotic prescriptions varied between 24% and 48% [13,14,17]. Thus, inappropriate prescribing of antibiotics in LRTIs is still a problem [18].

The present study lacks information on the duration of symptoms, which can affect the reliability of imaging results since consolidations sometimes are not visible until 48 h of symptom duration [19]. Typically, patients have already been experiencing illness for a few days prior to securing an appointment with their physician. Therefore, it is unlikely that short symptom duration would explain the lack of consolidation on imaging and therefore prompt the high prescription rate of antibiotics [13,20]. Likewise, if there was a suspicion of atypical pneumonia (e.g., Mycoplasma pneumonia), partly based on the information on low CRP levels, the chest imaging should probably have shown a new consolidation confirming the suspicion. Thus, antibiotic treatment would be inappropriate in these cases [21].

Low, but significantly different median CRP values were seen between those prescribed and those not prescribed antibiotics when imaging was negative (Table 2). This aligns with several previous studies [8,22,23]. CRP values above 50 mg/L are more frequently seen when a consolidation is present [23,24,25]. Rögnvaldsson et al. showed that patients with symptoms of pneumonia and negative CXR had significantly lower CRP levels and more frequently had detectable viruses in PCR testing than those with consolidation [22]. Thus, with viral etiology in the absence of a consolidation, the infection is probably another LRTI diagnosis, presumably acute bronchitis, which can also be suspected in our study. However, based on clinical examination, Lagerström et al. showed that 54% of patients diagnosed with pneumonia in primary care lacked consolidation on CXR despite positive etiological findings in half of the patients [21]. Although the duration of symptoms was not known in that study, this highlights how complex and challenging diagnosing pneumonia can be.

Many factors, including patient treatment expectations, may impact physician prescribing [26,27,28,29,30,31,32,33]. In this study, the co-morbidity rate was higher among those with antibiotics prescribed and negative imaging compared to those with antibiotics and positive imaging (Table 3). This may indicate that physicians are more eager to be active in treating patients with co-morbidity and do not want to risk severe complications of the infection or miss other causes of the symptoms. Those findings are in line with a qualitative study by Boiko et al., where some prescribers feared potential complications if they were not prescribed antibiotics (severe bacterial infections or sepsis) more than the possible adverse effects of prescribing antibiotics (antimicrobial resistance or complications such as diarrhea) [30]. Moreover, physicians’ perception of antimicrobial resistance affects their management of infections. In a Swedish qualitative study by Björkman et al., greater adherence to treatment guidelines for urinary tract infections (UTIs) was seen when the physician considered antibiotic resistance an important factor in managing UTIs [31]. Other studies have even described that physicians with high prescribing rates of antibiotics more often assess infections as severe and, to a greater extent, use diagnoses where antibiotic treatment is recommended compared to low prescribers, probably to legitimatize the prescription [27,28]. In this study, this could be an explanation for the high proportion of patients with the diagnosis of pneumonia despite negative imaging.

The use of PcV was low in this study, especially when co-morbidity was present. Inappropriate antibiotic prescribing for acute RTIs is common in primary care settings, reported between 48–76% in various studies [34,35,36,37]. Factors such as the physician’s lack of awareness of current guidelines, level of competence, the overall working load, and the patient’s experiences or wishes have been suggested as plausible explanations [26,27,28,29,30,31,32,33,38]. The physician’s suspicion of atypical genesis (e.g., Mycoplasma pneumoniae) or consideration of co-morbidity might have impacted the usage of PcV in this study. A documented allergy to penicillin limits treatment options. Unfortunately, information on antibiotics allergy was missing in our study. Resident physicians were more likely to prescribe antibiotics given the negative imaging compared to specialists or interns. We can only speculate on why, as the results in the literature diverge. In a Dutch study by Akkerman et al., the physician’s years of practice were the main factor correlating with high antibiotic prescribing rates, which is in contrast to a Norwegian register-based study where lower prescription rates were seen among experienced general practitioners [26,29]. One could imagine that residents may have been influenced by colleagues at hospital clinics while doing their vocational training. Patients admitted to hospitals are probably more severely ill and are therefore being managed differently than patients in primary care. One could also imagine that specialists supervise and support interns more closely than residents, thereby having a larger impact on prescribing habits in the former group. That could be a plausible explanation for the lower prescription rate among interns.

Mentoring and educating in primary care where residents are training may be valuable for healthy prescribing habits [27]. Our study indicates that this should be emphasized for the resident physicians since inadequate prescribing of antibiotics not only contributes to more antibiotic-resistant bacteria but also results in an increased number of consultations and increased demand for antibiotics in the future [39,40].

## 5. Conclusions

This register-based study showed that every other patient with acute bronchitis or pneumonia and negative chest imaging received antibiotics despite low median CRP values. Resident physicians were more inclined to prescribe antibiotics in the group with negative imaging. CRP levels were low on average but significantly higher among those treated with antibiotics. We did not find characteristics for antibiotic prescribing in LRTI cases with negative imaging. Our results indicate the need for further research exploring the pattern of antibiotic use in LRTIs and diagnosis of pneumonia among practitioners.

## Figures and Tables

**Figure 1 antibiotics-12-01165-f001:**
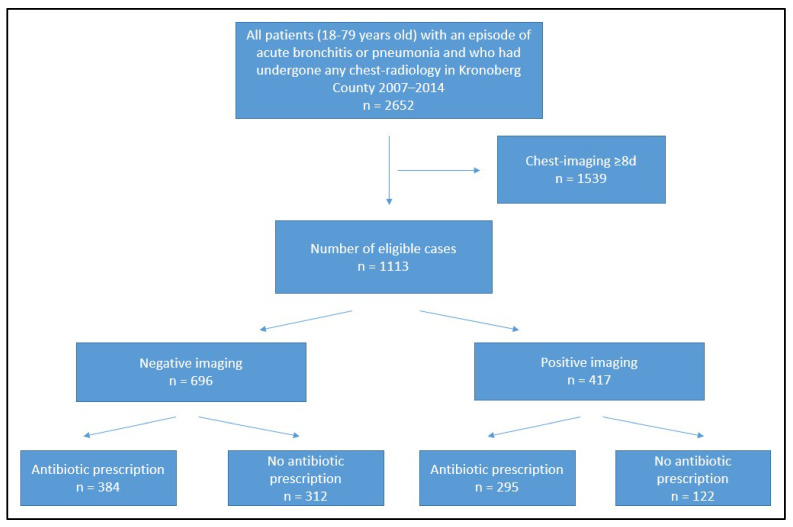
Study subjects screening, enrolment, and completion flow diagram.

**Figure 2 antibiotics-12-01165-f002:**
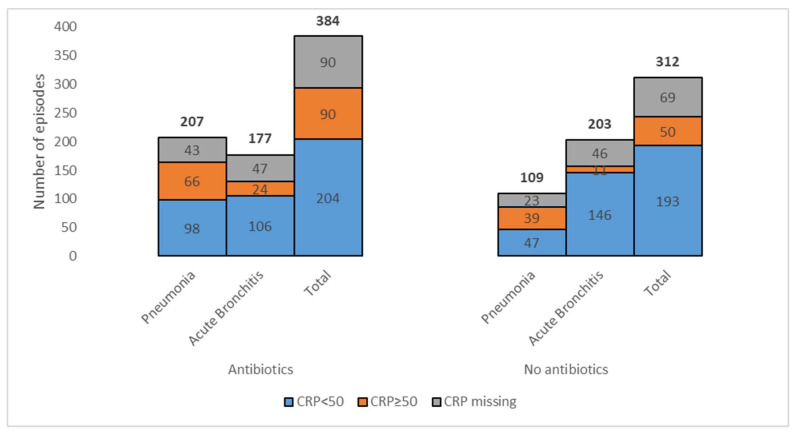
Distribution of C-reactive protein (CRP) with cut-off level of 50 mg/L in cases with negative chest imaging and the diagnosis pneumonia or acute bronchitis in which antibiotics were prescribed or not prescribed. The total number of cases independent of diagnosis is also shown for both groups. The precise numbers are shown on the figure (total number of cases for each bar is shown above the bars).

**Table 1 antibiotics-12-01165-t001:** Baseline demographics and clinical characteristics.

	Total (*n* = 1113)	Negative Imaging (*n* = 696)	Positive Imaging (*n* = 417)	*p* Value
Age (years), median (IQR)	58 (42;67)	59 (43;67)	57 (41;68)	0.53 ^a^
Men, *n* (%)	563 (51)	349 (50)	214 (51)	0.70
Co-morbidity, *n* (%)	493 (44)	335 (48)	158 (38)	**<0.001**
CRP mg/L, median (IQR)	30 (10;103)	13 (10;53)	79 (33;160)	**<0.001 ^a^**
Antibiotics, *n* (%)	679 (61)	384 (55)	295 (71)	**<0.001**
PcV, *n* (%)	195 (20)	90 (13)	105 (25)	**<0.001**
Diagnosis of pneumonia, *n* (%)	685 (62)	316 (45)	369 (89)	**<0.001**

Characteristics of primary care patients aged 18–79 years with an episode of acute bronchitis or pneumonia in which chest imaging was performed. CRP = C-reactive protein. IQR = Interquartile range. Pearson’s chi-square test was used for group comparison if not specified otherwise. *p* values < 0.05 were considered significant and are indicated in bold. ^a^ Mann–Whitney U test.

**Table 2 antibiotics-12-01165-t002:** Comparison of pneumonia and acute bronchitis cases with negative imaging and with or without antibiotics.

	Antibiotic Prescribed (*n* = 384)	No Antibiotic Prescribed (*n* = 312)	*p* Value
Age (years), median (IQR)	58 (43;67)	59 (43;67)	0.89 ^a^
Age ≥ 65 years, *n*	124	107	0.58
Men, *n* (%)	199 (52)	150 (48)	0.33
Co-morbidity, *n* (%)	184 (48)	151 (48)	0.90
CRP mg/L, median (IQR)	21 (10;60)	10 (10;37)	**<0.001 ^a^**
CRP > 50 mg/L, *n* (%)	90 (64)	50 (36)	**<0.05**
**Physician’s level of competence, *n* (%)**			**<0.001**
Specialist	229 (50)	228 (50)	
Resident physician	121 (67)	59 (33)	
Intern	34 (58)	25 (42)	

CRP = C-reactive protein. IQR = Interquartile range. CI = Confidence interval. OR = Odds ratio. Pearson’s chi-square test was used for group comparison if not specified otherwise. *p* values < 0.05 were considered significant and are indicated in bold. ^a^ Mann–Whitney U test.

**Table 3 antibiotics-12-01165-t003:** Comparison of cases of pneumonia and acute bronchitis with antibiotic prescription with positive or negative imaging.

	Negative Imaging (*n* = 384)	Positive Imaging (*n* = 295)	*p* Value
Age (years), median (IQR)	58 (43;67)	55 (37;67)	0.13
Men, *n* (%)	199 (52)	156 (53)	0.79
Co-morbidity, *n* (%)	184 (48)	108 (37)	**<0.05**
CRP mg/L, median (IQR)	21 (10;60)	72 (30;148)	**<0.001 ^a^**
First-line antibiotics, *n* (%)	90 (23)	105 (36)	**<0.001**
Diagnosis of pneumonia, *n* (%)	207 (54)	261 (89)	**<0.001**

CRP = C-reactive protein. IQR = Interquartile range. Pearson’s chi-square test was used for group comparison if not specified otherwise. *p* values < 0.05 were considered significant and are indicated in bold. ^a^ Mann–Whitney U test.

## Data Availability

The datasets generated and analyzed during the current study are not publicly available due to Swedish legislation (the Personal Data Act) but are available from the corresponding author on reasonable request.

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
