# Peer review of "Antibiotic Treatment for Lower Respiratory Tract Infections in Primary Care: A Register-Based Study Examining the Role of Radiographic Imaging"

_antibiotics, 2023, doi:10.3390/antibiotics12071165_

Round 1

Reviewer 1 Report

Reviewer comments and suggestions

The authors in this study explained the lower respiratory tract infections (LRTIs) with negative chest radiology that compares episodes where antibiotics are prescribed and not prescribed to find characteristics that could explain the prescription. 

Data were extracted from patients aged 18-79 years diagnosed with acute bronchitis or pneumonia and who had undergone any chest radiology between 2007 and 2014. 

The study observed that age, gender, and comorbidity did not differ between cases prescribed and not prescribed antibiotics. The study highlighted that when chest radiology was negative, no characteristics were found that could explain antibiotic prescription. 

Overall, the manuscript was well written. However, a few concerns/comments needed to be explained/modified. 

  1. Line 2 The title seems to be modified “treatment and radiology” word 
  2. Line 44 Could the author explain the references 3 and 4
  3. Line 61-62 is here any possible reason for this
  4. Table 2 Specialist (50) What does it mean or indicate
  5. Line 134-135 Did they have the data for this
  6. Line 163-165 What does it mean, could you please bifurcate the sentences for easy meaning
  7. Line 202-209 it would be nice if the authors could add the respective tables or figures where they discussed.
  8. Please check the guidelines of MDPI, it seems that the authors need to modify all the references.

Reviewer 2 Report

Manuscript ID: antibiotics-2456749
Title: Antibiotic treatment and radiology in lower respiratory tract infections in primary care - a register-based study.
General comments: Not sure if the title should be reversed to: Treatment of lower respiratory tract infections with antibiotics based on radiographic imaging in primary care - a register-based study.  
There are too many short paragraphs and can be combined to eliminate extra wording.
Maybe changing "episodes" to "cases".
Based on the author guidelines, the article should have the following formatting.  Title page, abstract, introduction, Materials and methods, results, discussions, etc.  Please consider aligning your article headings/titles based on the guidelines.     
Not sure if we can say medical residents in training rather than "Resident Physicians".  Just a suggestion.
ABSTRACT:
Line #18 - May start the sentence when imaging (i.e., chest x-ray or CT) and then you should consider using the word imaging throughout the manuscript.  It should be clear to your readership.
Line #19 - Just a minor comment after pneumonia, many patients/subjects are prescribed antibiotics despite the absence of pneumonia signs/symptoms.

Line #23 - You have used some abbreviations here; may consider abbreviating EMR after you have spelled it out.
Line #25 - just shorten it who had any chest radiologic imaging between 2007 - 2014.
Line #26 - Of 696 cases without evidence of infiltrates on imaging, 55% were prescribed antibiotics.
Line #27 - did not differ between those with or without antibiotics.   Change more often to more frequently.
Line 31 Line #32 - It is unclear what features/manifestations prompted the antibiotic prescribing in those with negative imaging indicating overuse of antibiotics for pneumonia and LRTIs.
 INTRODUCTION:
Line #38 - Line #40 - Consider changing: Although important, it can be challenging to differentiate .....
Line #40 - Change "lethal" just plain "serious".  This is how infection is described often as "serious."
Line #41 - I think you should insert "should be treated".   Delete "therefore".
Line #41 - Just limit to: In Sweden, phenoxymethylpenicillin is the first line therapy for pneumonia.   
Line # 43 - Line #44 - Consider combining with the first paragraph since it is still related to RTIs, and this is a very short paragraph.  
Line #45 - Line #48 -Consider changing "development of ...." to "emergence of antimicrobial resistance" and is increasing worldwide leading to global health causing approximately 700,00 deaths annually.  Therefore, addressing the inappropriate antibiotic prescribing can be addressed by education, training, experts of infectious disease, better diagnostic tools, and use of resistance profiles.   
 Line #50 - Line #51 - Not sure if this sentence is needed; you have rule in/out and set-up a rule (too many rules).  I thought almost everyone follows infectious disease guidelines to some extent.  Delete
Line #51 - Line #54 - Start the sentence with: Currently the gold standard for pneumonia diagnosis in Chest x-ray (CXR).  However, imaging access in primary care is almost always .....  
Line #56 - Line #58 - Should this be added to above when you discuss the gold standard the X-ray and examination?  In most cases, the gold standard for pneumonia diagnosis is a chest x-ray, physical examination, blood and sputum test.  However, this approach often results in ....
Line #64 - You say the aim of this study was... then later Line #69 you have secondary.  Should you start with: the Primary objective of this study was to look at the pattern (better fit) vs. prevalence of....The secondary outcome was to compare the two groups: antibiotics with negative imaging vs. antibiotics and positive imaging.   
RESULTS:
Line #76 - Line #77 - Seems a little confusing, Radiology was performed and then CXR was carried out in one????
Line # 77 - Reverse to: No consolidation was seen in 696 cases based on the radiographic studies.
Line #81 - Consider revising to keep the less significant items together and move the ischemic heart disease and asthma the end.  Here is a suggestion:  Age and gender differences were not significant between the positive and negative imaging group.  The proportion of COPD......Whereas, the comorbidities such as ischemic heart disease and asthma was more common in the negative radiologic/imaging group.
Line #86 - Line #87 - Consider combining these two short paragraphs.   (line# 81 - Line# 89).  CRP was available in 80% of cases and microbiological testing in 10%.
Line #101 - Line #102 - Reverse to: Antibiotics were prescribed to 55% of subjects with negative imaging.  Age, gender, and comorbidity did not impact the antibiotic prescribing pattern.  (data not shown)!  Why Not?
Line #104 - Consider using when CRP > 50 mg/L.
Page #5
Not sure why the line numbering was interrupted: I will do my best to identify the section:
2.2 Antibiotic prescribing and negative imaging
Paragraph #3 (page 5) -Revise to:  A total of 23% received antibiotics despite the negative imaging.  An underlying comorbidity did not impact the antibiotic prescribing (p=0.086) and no difference was seen with regards to physician's level of competence (P=0.65).  
2.3 All episodes with antibiotic prescribing
I would suggest moving (table 3) to the end of this paragraph since later you discuss the CRP values as well.
Second Paragraph - Revise to: Fewer antibiotics were prescribed in those with negative imaging.  However, if a comorbidity was present the extent of antibiotic prescribing was the same in both groups.
Not sure if you need to repeat "first line" over and over, just say antibiotic(s).
DISCUSSIONS:
Line #131 - Delete the word "often" not needed,
Line #132 - Consider changing "minor" to "Small"; also, revise the sentence to: there was a small difference in the characteristics of those who received antibiotics vs. those who did not.
Line #136 - 139 - Not sure why we are repeating this again.
Line #142 - Line #146 - I would suggest starting the paragraph with: The strengths of our study was the large sample which included data from the entire county and all PHCCs (define this, not seen the abbreviation), and out of hour offices (not sure what you mean by this?  Is this supposed to say after hours? Also, what do you mean by "no online doctors"? does this mean "virtual visits"?   Please clean and clarify this paragraph.  
Then go on with the limitations:  
Our study had few limitations which may have impacted the results.  Data on smoking, concomitant medications, allergies, and immune system (competent or compromised) were missing.  In addition, the timing of antibiotic in relation to radiologic studies.
Line #153 Line #155 - Not sure what we are trying to report?  Is this only the electronic prescriptions were included, the rest of the sentence needs clarifying.
Line #157 -
Line #161 - should this be added to the limitations as well?
Line #163 - are we still talking about present (your study)? Is so, then delete "in this study" just combine with "further" the short duration of symptoms would ....
Line #164 - Not sure if "motivate" is the correct terminology, please consider changing to "prompt."   
Line #183 - Delete "can have an" just say may impact physician's prescribing pattern.
Line #196 - Line #201 - Not sure what you are trying to say here and why is it important?  
Line #202 - Delete "first line", just say the use of antibiotics in this study was low,
Line #203 - Abbreviate RTI, has already been spelled out.
Line #204 - reported between 48-76% in various studies.
Line #207 - What do you mean by atypical genesis ....
Line #210 - Delete "Material", just say the allergy information was missing in our study; I am very surprised, and do you guys try to verify if this was a true allergy or just a rash?  Often these allergies are not significant!!! Just wondering.
Line #210 - Line #211 - Revise to: resident physicians were more likely to prescribe antibiotics given the negative imaging compared to specialist or interns (just finish here, no need for extra wording).
Line #216 - Spell out GP first before abbreviation.  I am assuming it is General Practitioner.
Line #221 - Should be prescribing habits.
Line # 223 -Chang the start of the sentence to: mentoring and educating in primary care where residents are training may be valuable for healthy prescribing habits.
MATERIALS AND METHODS:
Line #231 - Line #241 - I will try to suggest some adjustment:  This retrospective registered-based study used data from the KIDPC which included information on subjects with infectious disease diagnosis and the prescribed antibiotics in the primary care of ...
PHCCs was abbreviated first in Line #144, you should spell it out first then use the abbreviation here in Line #234.
Line #235 - Again, please clarify "out-of-hours" offices as mentioned previously.
Line #240 - You mention at one occasion in 2015 - what do you mean by this?
Line # 243 - Just start with: We included patients 18 - 79 years of age with acute bronchitis or pneumonia and chest radiology within 7-days after the first consultation.  Delete the Kronoberg County.
Line #252 - Consider revising the sentence to: We did not include patients over 79 years of age due to the lack of EMR connected prescription history.  
Line #254 - Line #259 - Not sure what you are trying to say by this sentence Contacts occurring.....Then you say 23 patients were found to have more than one episode, is this pneumonia episode? It is very confusing.  Please consider clarifying.  Clean up the last sentence to say: The chest radiologic results were divided into two groups: negative/normal exam (no signs of pneumonia vs. positive (pneumonia signs visible).
CONCLUSIONS:
Line #272 - Delete "especially", just start with Resident physicians were more inclined ....
Line #273 - Delete the "with negative radiology" you have already mentioned along the low CRP.   Delete this sentence totally.  Then go on with: We did not find characteristics (not sure if you can use hallmarks, indication, or criteria here) for antibiotic prescribing in LRTIs cases with negative imaging.  Finish the paragraph with: Our results indicate the need for further research exploring the pattern of antibiotic use in LRTIs and diagnosis of pneumonia among practitioners.   
The conclusions should be 3-4 short sentences.   Sorry, just made some suggestions for you to consider.
Figure #1
Change the title to: Study subjects screening, enrolment, and completion flow diagram.
Table #1  
Change the title to Baseline Demographics and clinical characteristics.  You have other things listed on the table.  
Table #2
Consider changing the table title to: Comparison of pneumonia and acute bronchitis cases with negative imaging and with or without antibiotics.

I hope my comments will be well received,.  There are many issues with the manuscript including the overall formatting, I have suggested some changes including few vocabulary ideas for improvement.  Few sections can be eliminated or combined; too many repetitive sections with poor flow of the concept.   

Reviewer 3 Report

I appreciate the opportunity to review Antibiotic treatment and radiology in lower respiratory tract infections in primary care – a register-based study. Although the manuscript generally reads well, closer inspection has raised some concerns that should be addressed by the authors.

1. The demographical data are incomplete. For instance, data including body mass index is missing

2.  A multi regression LOGISTIC method must be applied in order to obtain the odds ratio for positive radiology and antibiotics.

3. Please better describe the situation in Sweden with regards to the use of antibiotics.

4. I am a little bit concerned about the association of antibiotics and positive radiology. It is not the sole marker for concluding that "When chest radiology was negative, no characteristics were found that explained antibiotic prescription. This could indicate an overuse of antibiotics in patients with LRTI."  This is a hard conclusion. Please modify to lower the impact of this statement.

Careful re-reading by the authors is recommended to take care of the minor editorial blemishes including grammar, punctuation, spelling, space, misplaced words and improvement of overall readability. It is recommended that the authors have their manuscript checked by an English language native speaker.

Round 2

Reviewer 3 Report

The responses to my questions raised in my initial review are adequately addressed. I have no further comments.